# Development of pacemaker properties and rhythmogenic mechanisms in the mouse embryonic respiratory network

**Marc Chevalier[1], Natalia Toporikova[2], John Simmers[1], Muriel Thoby-Brisson[1]***

[1]Institut de Neurosciences Cognitives et Intégratives d'Aquitaine, CNRS UMR 5287, Université de Bordeaux, Bordeaux, France; [2]Department of Biology, Washington and Lee University, Lexington, United States

**Abstract** Breathing is a vital rhythmic behavior generated by hindbrain neuronal circuitry, including the preBötzinger complex network (preBötC) that controls inspiration. The emergence of preBötC network activity during prenatal development has been described, but little is known regarding inspiratory neurons expressing pacemaker properties at embryonic stages. Here, we combined calcium imaging and electrophysiological recordings in mouse embryo brainstem slices together with computational modeling to reveal the existence of heterogeneous pacemaker oscillatory properties relying on distinct combinations of burst-generating $I_{NaP}$ and $I_{CAN}$ conductances. The respective proportion of the different inspiratory pacemaker subtypes changes during prenatal development. Concomitantly, network rhythmogenesis switches from a purely $I_{NaP}$/$I_{CAN}$-dependent mechanism at E16.5 to a combined pacemaker/network-driven process at E18.5. Our results provide the first description of pacemaker bursting properties in embryonic preBötC neurons and indicate that network rhythmogenesis undergoes important changes during prenatal development through alterations in both circuit properties and the biophysical characteristics of pacemaker neurons.

**\*For correspondence:** muriel. thoby-brisson@u-bordeaux.fr

**Competing interests:** The authors declare that no competing interests exist.

## Introduction

Rhythmic motor activities are generated and controlled by neuronal networks organized as central pattern generators (CPG) (*Marder and Bucher, 2001*; *Harris-Warrick, 2010*). Considerable data accumulated over the last decades from both invertebrate and vertebrate models have established the general mechanistic principle that rhythmogenesis relies on an interplay between intrinsic neuronal membrane properties and intercellular synaptic connectivity. Two main processes that may operate in varying combinations underlie motor rhythm generation: (i) the CPG network in question contains endogenously oscillatory neurons, so-called pacemakers, which drive the wider circuit cell population, and/or (ii), the rhythm emerges from the pattern of synaptic connections within the network. In addition to these intrinsic rhythmogenic mechanisms, the dynamics of network function can be conferred by extrinsic neuromodulatory actions. By acting on the membrane properties of constitutive neurons or their synaptic interconnections, modulators can ensure the operational plasticity that enables network motor output to remain adapted to organismal needs and changing environmental conditions (for reviews, see *Harris-Warrick, 2011*; *Marder, 2011*; *Marder et al., 2014*).

A major physiological function of such a CPG is breathing. Respiratory movements are driven by rhythmic motor activity generated by neuronal circuits located in the brainstem. The respiratory rhythm generator is composed of two interacting CPG circuits distributed bilaterally in the ventral part of the medulla. The first is the parafacial respiratory group (RTN/pFRG; *Onimaru and Homma, 2003*) that appears to generate preinspiratory activity in neonates in vitro, active expiration in adults

**eLife digest** Babies need to start breathing immediately when they are born. Researchers have detected rhythmic movements in the fetus that are related to breathing, which supports the idea that the nervous system circuits needed for breathing are still being established and refined shortly before birth. Neurons in a part of the brain called the brainstem control the muscles that generate the movements required for breathing. These neurons are organized into groups, with each group forming an independent network that can carry information in the form of electrical signals. However, it is not clear how these networks form and operate before birth.

Chevalier et al. tracked electrical activity in slices of brainstems from mouse embryos. The experiments show that these embryos already have 'pacemaker' neurons that can drive rhythmic activity in the networks of neurons related to breathing. There are several types of pacemaker neurons that produce different patterns of electrical firing. The amount of each type of pacemaker in the brainstem changes in the later stages of the pregnancy.

The experiments also show that the way in which pacemaker neurons control the networks of breathing-related neurons changes as the embryo develops. Early on in development, pacemaker neurons play an essential role in generating rhythms in the other neurons. However, in older embryos, the connections between each neuron in the breathing network become more important. Further work is now needed to map out the exact sequence of events in embryos that allow mice to breathe as soon as they are born. This could help us to develop therapies for human babies that are born with breathing difficulties.

and plays a prominent role in central chemosensitivity (*Guyenet and Bayliss, 2015*). The second network is the preBötzinger complex (preBötC; *Smith et al., 1991*) which has now been established to be both sufficient and necessary for generating the inspiratory phase of respiration (*Smith et al., 1991*; *Gray et al., 2001*; *McKay et al., 2005*; *Tan et al., 2008*; *Bouvier et al., 2010*). The excitatory glutamatergic preBötC network contains ~800 neurons, some of which (≤15%) in neonatal mouse exhibit intrinsic pacemaker properties (*Koshiya and Smith, 1999*; *Thoby-Brisson and Ramirez, 2001*; *Pena et al., 2004*). To date, a leading hypothesis, inscribed in the 'group pacemaker hypothesis', proposes that the rodent postnatal respiratory rhythm derives from an interaction between membrane properties (including pacemaker cellular properties) and synaptic coupling (*Rekling and Feldman, 1998*; *Feldman and Del Negro, 2006*; *Feldman et al., 2013*).

It has been shown in rodents that the preBötzinger complex becomes functional during the last third of gestation (*Pagliardini et al., 2003*; *Thoby-Brisson et al., 2005*). Already at early embryonic stages, glutamatergic synaptic signaling is required for preBötC network output (*Thoby-Brisson et al., 2005*; *Wallen-Mackenzie et al., 2006*), although the presence of embryonic inspiratory neurons endowed with intrinsic bursting properties has only been inferred (*Thoby-Brisson et al., 2005*; *Bouvier et al., 2008*). Therefore, the aim of this study was to establish the presence and biophysical characteristics of pacemaker neurons in mouse embryonic preBötC circuitry in order to understand their development and contribution to respiratory network activity in the critical period immediately prior to birth.

## Results

### Heterogeneous discharge patterns of embryonic inspiratory pacemaker neurons

To identify pacemaker neurons in preBötC respiratory circuitry of mouse embryos between E16.5 and E18.5, we combined electrophysiological recordings of population rhythmic activity on one side with individual cell calcium imaging on the contralateral side of brainstem slice preparations (*Figure 1A*). For this, slices were previously incubated en bloc with the Calcium Green 1-AM indicator, allowing fluorescence fluctuations due to somatic $Ca^{2+}$ fluxes resulting from spontaneous impulse burst generation to be monitored (see Materials and methods). Initially, rhythmic fluorescent changes in cells occurring in phase with the population electrical activity allowed the localization of

inspiratory neuron somata (*Figure 1A*, right and *Figure 1B*). We identified an endogenous pacemaker neuron by its ability to produce spontaneous membrane potential oscillation and rhythmic action potential burst discharge even in synaptic isolation from its network partners (*Koshiya and Smith, 1999*). Accordingly, neurons expressing fluorescence fluctuations in time with fictive inspiration in control conditions were classified as pacemakers if they remained rhythmically active during subsequent exposure to a cocktail of agents known to block chemical synaptic transmission in the preBötC network (see Material and methods). Under such conditions of synaptic blockade, network electrical activity ceased as did rhythmic fluorescent changes in most of the previously identified inspiratory neurons (*Figure 1C*, black traces). However, a small proportion of monitored cells continued to express spontaneous fluorescence fluctuations at unrelated frequencies (*Figure 1C*, red traces). These specific neurons were therefore considered to be pacemaker cells and were targeted for patch-clamp recording.

Of the 84 pacemaker neurons identified in 69 embryo slice preparations, three distinct types of discharge pattern were observed that differed in the characteristics of the spontaneous depolarizing waveforms - or drive potentials (DPs) – that underlie their intrinsic bursting activities. In a first group (n = 23), the cells expressed long-lasting plateau-like DPs with action potentials occurring at the beginning of the plateau followed by a depolarization block during which the neuron remained at a depolarized membrane potential without further spike generation prior to a spontaneous return to resting potential (*Figure 2A*). The mean amplitude of such square-wave DPs was 30.5 ± 6.6 mV (343 burst cycles measured from 12 neurons; *Figure 2D$_1$*, left) and their mean duration was 2.9 ± 0.1s (*Figure 2D$_1$*, right). The mean membrane potential of these neurons measured between their DPs was −52.7 ± 0.7 mV. In a second group (n = 45), the pacemaker neurons generated short-lasting oscillatory DPs and associated bursts with depolarizing amplitude and duration means of 12.8 ± 3.8 mV and 0.79 ± 0.01 s, respectively (1296 bursts measured from 23 neurons; *Figure 2B, D$_2$*). Their mean membrane potential between bursts was −49.2 ± 0.5 mV. In the third group (n = 16), cells expressed a mixture of long- and short-lasting DPs, which overall had amplitude and duration means of 14.5 ± 7.5 mV and 1.74 ± 0.08 s, respectively (739 bursts measured from 15 neurons; *Figure 2C, D$_3$*). The mean inter-burst membrane potential of the mixed phenotype was −49.1 ± 0.7 mV. The durations of the drive potentials were statistically different between the three groups (Mann-Whitney test, p<0.001; *Figure 2D$_4$*, right), while the mean DP amplitude of plateauing pacemakers was significantly larger compared to the amplitudes of two other types (p< 0.001), which themselves were not significantly different (*Figure 2D$_4$*, left). However, for the mixed pacemaker phenotype, when we discriminated between the two types of bursting (light green bars in *Figure 2D$_4$*), the mean values for both DP amplitude and duration were comparable to those of the separate plateauing and oscillatory bursters. Moreover, as evident in the DP amplitude vs duration relationship of *Figure 2E* (which note was plotted exclusively from measurements of long-term single cell recordings; see Material and methods), the values for the mixed phenotype were bimodally distributed, with its shorter bursts lying in the range for oscillatory only pacemaker neurons and its longer bursts overlapping values for plateauing only cells. Finally, the mean resting membrane potential values were statistically different between both plateauing and oscillatory neurons (t-test, p<0.001) and between plateauing and mixed pacemaker cells (t-test, p<0.002). Thus, together these data show that the respiratory network at embryonic stages between E16.5 and E18.5 already contains a subpopulation of pacemaker neurons and that these cells are endowed with heterogeneous burst-generating intrinsic properties.

## Membrane conductances underlying the different discharge patterns

We next investigated the membrane properties involved in the pacemaker activities of these embryonic inspiratory neurons. *I-V* curves obtained from isolated plateauing (n = 6) and oscillatory (n = 14) pacemaker neurons expressed a non-linear deviation at hyperpolarized membrane potentials, indicating the activation of voltage-dependent membrane conductances (*Figure 3A*). Consistent also with the voltage-dependence of an endogenous pacemaker mechanism, membrane potential depolarization with current injection caused a cycle frequency increase in both types of bursting neuron, with the rates of plateauing and oscillatory bursting increasing as a function of injected current intensity (*Figure 3B,C*). It is also noteworthy, however, that the *I-V* relationships of the two pacemaker phenotypes were statistically different (*t*-test; p<0.001) in the range of more depolarized membrane potential levels between −20 mV and 0 mV, thus indicating differences in properties that govern

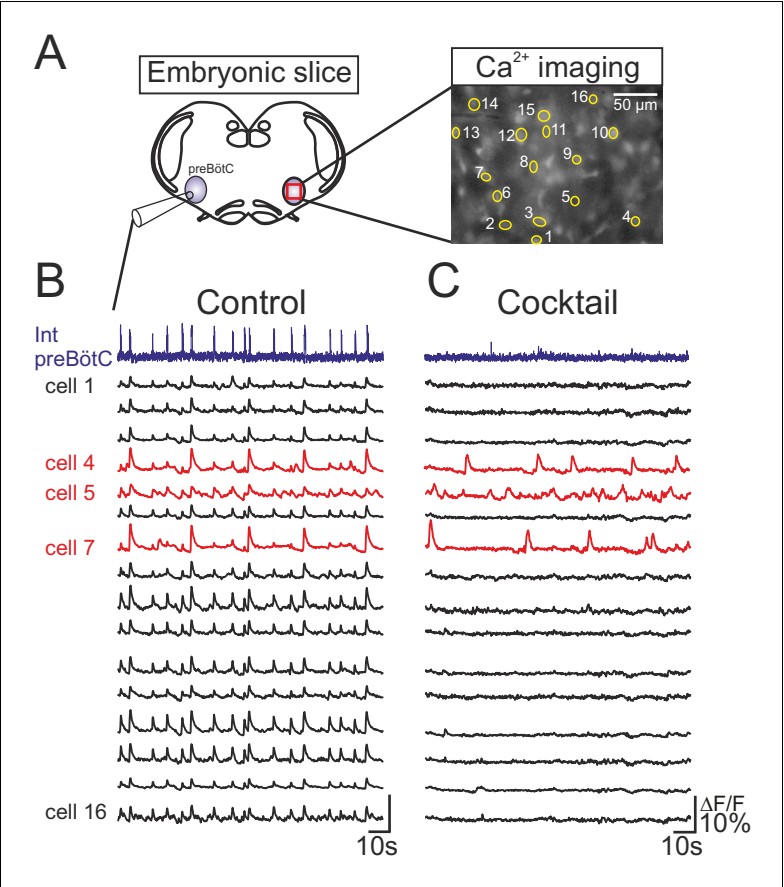

**Figure 1.** Functional isolation of embryonic inspiratory pacemaker neurons. (**A**) left, Schematic representation of an in vitro embryonic slice preparation used for making electrophysiological macroelectrode recording of preBötC network activity on one side simultaneously with calcium imaging of the contralateral inspiratory network (red square). Right, Image of fluorescence expression by a $Ca^{2+}$-dye loaded slice obtained with a 40X objective. Numbered yellow circles indicate rhythmically active neurons. (**B**) Simultaneous recordings of inspiratory population electrical activity (integrated trace in violet, Int preBötC) and calcium transients ($\Delta F/F$, black and red traces) in 16 individual neurons (numbered 1 to 16). All the cells displayed fluorescent changes synchronized to rhythmic electrical activity recorded on the contralateral side in control conditions. (**C**) After blockade of chemical synapses by a bath- applied cocktail of CNQX (20 µM), AP5 (10 µM), strychnine (1 µM) and bicuculline (10 µM), neurons that remained rhythmically active (red traces) were considered to be pacemakers while those falling silent were considered as non-pacemaker cells (black traces).

their membrane excitability. No difference (*t*-test, p= 0.7) was found in the membrane input resistance measured around resting potential for plateau-like (1756 ± 667 MΩ) and burst-like pacemaker neurons (1574 ± 973 MΩ).

In order to identify the major membrane conductances contributing to the intrinsic burst activity of these embryonic pacemakers, we blocked the persistent sodium current ($I_{NaP}$) and the calcium-activated non-specific cationic conductance ($I_{CAN}$) that are known to be widely implicated in the rhythmogenic mechanisms of mammalian motor circuits (***van Drongelen et al., 2006***; ***Zhong et al., 2007***; ***Tazerart et al., 2008***; ***Tsuruyama et al., 2013***), including the preBötC respiratory network of the postnatal mouse (***Thoby-Brisson and Ramirez, 2001***; ***Pena et al., 2004***; ***Del Negro et al., 2005***; ***Paton et al., 2006***; ***Pace et al., 2007b***, ***2007a***). We therefore bath-applied Riluzole (Ril; 10 µM) a blocker of $I_{NaP}$ (***Urbani and Belluzzi, 2000***), and Flufenamic Acid (FFA; 50 µM) a blocker of $I_{CAN}$ (***Guinamard et al., 2004***) onto synaptically-isolated, patch-clamp recorded neurons. Note that since to our knowledge none of these two drugs are fully washable, we applied FFA and Ril alone, or in combination, but never sequentially on a given slice. Out of the 10 embryonic preBötC neurons

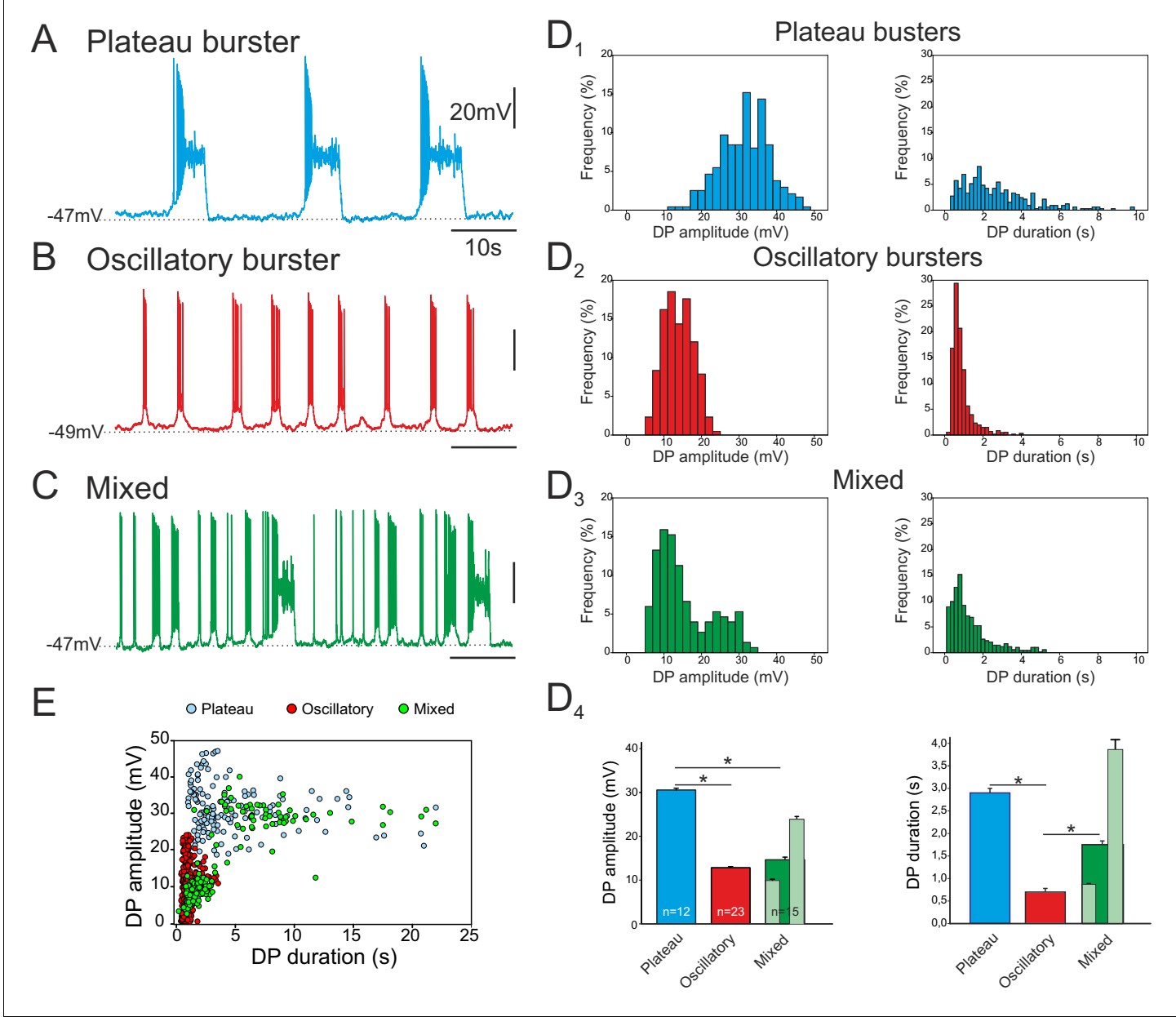

**Figure 2.** Different discharge patterns expressed by embryonic inspiratory pacemaker neurons. Patch-clamp recordings of individual pacemaker neurons from three different preparations expressing spontaneous plateau (A), oscillatory (B) or mixed plateau/oscillatory (C) burst firing patterns. (D) Histograms showing frequency distributions of the burst-generating drive potential (DP) amplitude ($D_{1-3}$, left column) and duration ($D_{1-3}$, right column) for the three types of pacemaker neurons. $D_4$: Histograms representing mean DP amplitude ± SEM (left) and mean DP duration ± SEM (right) for pacemaker neurons expressing plateau-like (blue bars), burst-like (red bars) or mixed (green bars; light green bars correspond to short-lasting and long-lasting bursts when grouped separately) activity. Asterisks indicate significant differences (Mann-Whitney test; $p<0.001$), while numbers of neurons analyzed for each pacemaker phenotype are indicated in the corresponding bar. (E) Distribution plot for DP duration *vs* DP amplitude measured for neurons recorded for 8 to 10 min (blue, plateau bursting pacemakers (n = 8); red, oscillatory bursting pacemakers (n = 5); green, mixed pacemakers (n = 5). Note that the latter expressed two types of burst duration/amplitude relationship that overlapped with either the purely plateau or oscillatory bursters.

identified as plateauing pacemakers, the bursting activity of 8 of these was completely blocked in the presence of 50 µM FFA applied either alone (n = 4; *Figure 4A*, right panel) or in co-application with 10 µM Ril, which by itself was ineffective in blocking bursting activity (n = 4, *Figure 4A*, left panel). For the remaining two plateauing cells, bursting activity was blocked in the presence of Ril

alone (n = 2). These findings therefore suggest that pacemaker activity of the plateauing inspiratory neurons relies on a combination of both $I_{NaP}$ and $I_{CAN}$, with a predominant role played by the latter.

For the vast majority (13/17) of the oscillatory bursting neurons tested, pacemaker activity was blocked in the presence of 10 µM Ril alone (n = 8; data not shown) or when co-applied with FFA (n = 5; *Figure 4B*). In the remaining four neurons, bursting was blocked by 50 µM FFA alone. For pacemaker neurons exhibiting a mixed DP pattern (n = 10), bursting activity was differentially sensitive to FFA and Ril, with plateau-like bursts being suppressed by FFA while short-lasting oscillatory bursts were blocked by Ril (*Figure 4C*). Thus, the heterogeneous patterns of discharge in pacemaker neurons of the embryonic preBötC network within the period from E16.5 to E18.5 appear to be associated with different combinations of membrane conductances contributing to underlying drive potential activity: $I_{CAN}$ plays a major role for plateau pacemakers, $I_{NaP}$ predominates in oscillatory pacemakers, whereas the relative conductance contribution in the mixed pacemaker phenotype is intermediate between these two.

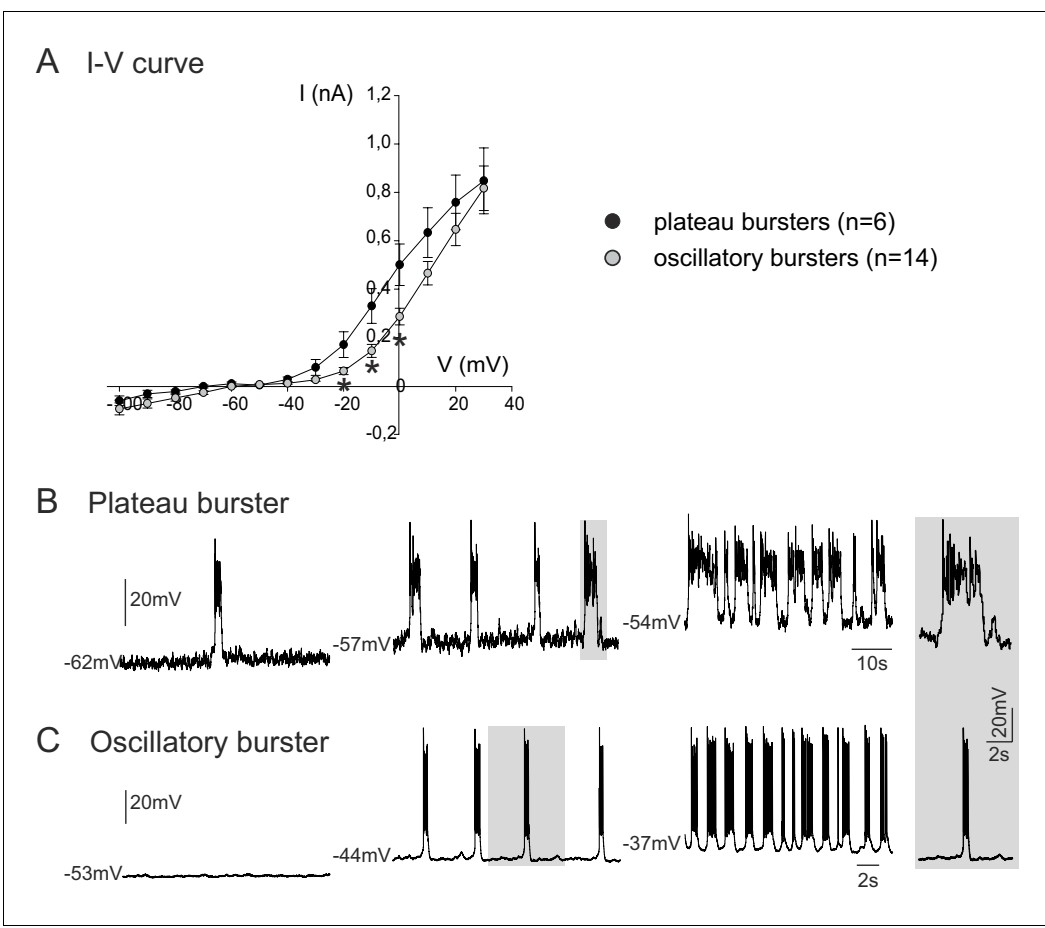

**Figure 3.** Voltage-dependence of pacemaker neuron properties. (A) Current-voltage (*I-V*) relationships for plateau (black circles) and oscillatory bursting (open circles) pacemaker neurons. Individual circles indicate mean (± SEM) current response to an injected step-change voltage pulse; n = number of neurons analyzed for each pacemaker phenotype. Asterisks indicate statistically different values (t-test; p<0.001). (B,C) Patch-clamp recordings of plateau (B) and oscillatory bursting (C) pacemaker neurons displaying cycle frequencies that varied with membrane voltage when held at different indicated membrane potential levels with continuous depolarizing current injection. Right inset: A single burst (indicated by shading) in each pacemaker neuron subtype displayed at the same time scale for burst duration comparison.

## Developmental changes in composition of the preBötC pacemaker population

We next determined whether the distinct types of pacemakers (plateauing, oscillatory; $I_{NaP}$-dependent, $I_{CAN}$-dependent) occur similarly throughout the prenatal period tested. Initially, pacemaker neurons at E16.5 (n = 27) and E18.5 (n = 35) were classified according to their discharge patterns obtained with single cell patch-clamp recordings at the two embryonic stages. Plateau and oscillatory pacemakers were found to be almost equally present at E16.5, whereas oscillatory bursters were predominant (65.7%, n = 23/35) at E18.5 (*Figure 5A*). At this later stage furthermore, the activity of only 8% (n = 3/35) of neurons was exclusively plateau-like, with the remaining 25.5% (n = 9/35) of cells expressing a mixed oscillatory/plateauing phenotype. These proportions of the different pacemaker cell types were significantly different between the two embryonic stages (chi-square test, p<0.001).

To discriminate between $I_{NaP}$-dependent and $I_{CAN}$-dependent pacemaker neuron activities in the two age groups, we compared sensitivity to Ril exposure in multiple-cell calcium imaging recordings at E16.5 and E18.5 (*Figure 5B*). It should be noted here that we were unable to use complementary FFA application in these imaging experiments because the $I_{CAN}$ blocker's effect on overall cell calcium signaling leads to a reduction of any initially detectable fluorescent changes, thereby preventing unequivocal interpretation. Of 32 pacemaker neurons identified at E16.5, 19 were sensitive to Ril exposure (*Figure 5C*, left histogram) indicating that the preBötC pacemaker population contains significant proportions (ratio 60:40) of both $I_{NaP}$ and presumed $I_{CAN}$ at this earlier embryonic age. In contrast, at E18.5, pacemaker neurons were found to be predominantly sensitive to Ril (22/27 neurons; *Figure 5C*, right), indicating that $I_{NaP}$ is now the effective rhythmogenic mechanism in the majority of the pacemaker population at this older stage. The proportions of $I_{NaP}$- and $I_{CAN}$-dependent pacemaker types found at the two developmental stages examined, despite not being statistically different in our imaging data (chi-square, p= 0.1), nevertheless, expressed a tendency that corresponded to our findings from patch-clamp recordings (see *Figure 5A*). Therefore, altogether these data support the conclusion that at E16.5 both types of pacemakers (riluzole sensitive $I_{NaP}$-dependent and riluzole-insensitive $I_{CAN}$-dependent) are present in comparable proportions, while 2 days later in prenatal development, the same preBötC cell subset is mainly comprised of $I_{NaP}$-dependent pacemaker neurons.

## The balance between $I_{NaP}$ and $I_{CAN}$ can determine the pacemaker discharge pattern

In principle, two developmental processes (or their combination) could underlie the different distributions of inspiratory pacemaker subtypes at different embryonic ages. One possibility is that $I_{CAN}$-dependent pacemakers progressively switch to an $I_{NaP}$-dependence as their membrane properties mature, thus involving a modification in the balance between the two conductances at the single cell level. Another alternative possibility is that $I_{CAN}$-dependent pacemaker neurons disappear progressively with embryo maturation and are replaced by a distinct $I_{NaP}$-dependent pacemaker population that emerges over the same period. To directly test the former possibility (also see below), a computational approach was used to assess the outcome of changing the proportion of $I_{NaP}$ and $I_{CAN}$ distribution on the discharge of a previously reported model inspiratory neuron (*Table 1*, *Figure 6A*; *Toporikova et al., 2015*). In a first trial, the total persistent sodium conductance ($g_{NaP}$) of the artificial cell's membrane was held constant at 2.5 nS while the total conductance for $I_{CAN}$ channels ($g_{CAN}$) was made variable. With $g_{CAN}$ at 2.5 nS, the model neuron produced repetitive, long-duration oscillations and burst discharge events (*Figure 6B*) that strongly resembled the plateau-like rhythmic activity recorded from biological pacemaker neurons (c.f., *Figure 2A*). In contrast, when $g_{CAN}$ was reduced to 0 nS, higher frequency short-lasting bursts were now produced (*Figure 6D*) in a manner that was strikingly similar to the oscillatory pacemaker phenotype observed in vitro (c.f., *Figure 2B*). On the other hand, a hybrid model activity pattern (*Figure 6C*) that closely resembled the mixed biological pacemaker phenotype (c.f., *Figure 2C*) was observed when $g_{CAN}$ was set at an intermediate conductance value of 1 nS.

Comparable results were also found when the opposite simulation paradigm was applied, whereby $g_{CAN}$ was held constant (at 1.5 nS) while $g_{NaP}$ was now varied between 0 and 5 nS (data not shown). Here again, as the proportion of artificial $I_{NaP}$ conductance relative to $g_{CAN}$ was increased,

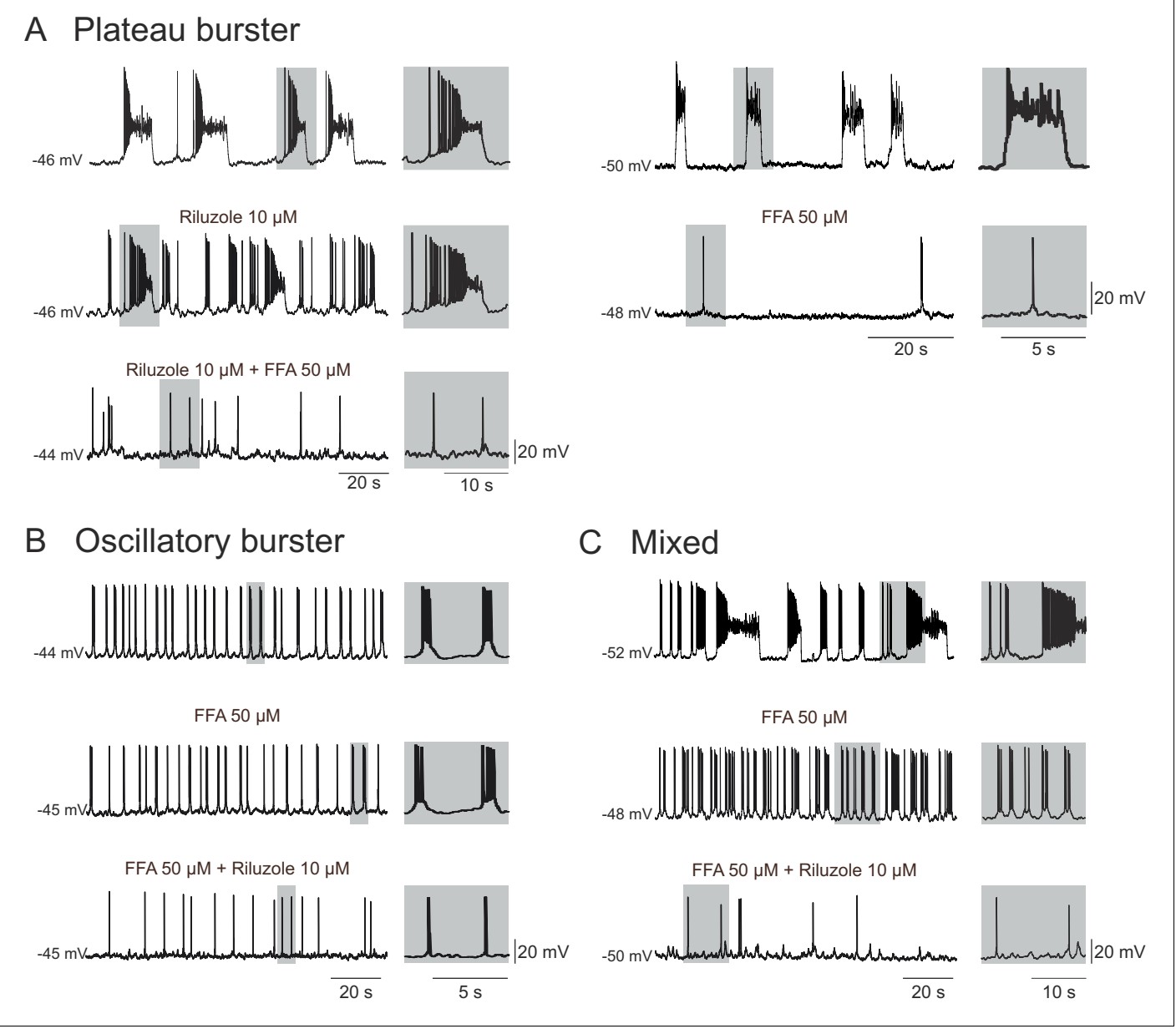

**Figure 4.** Membrane conductances underlying the different discharge patterns. Patch-clamp recordings of plateau (**A**), oscillatory (**B**) and mixed (**C**) bursting pacemaker neurons in control conditions (top traces), in the presence of either 10 µM Ril or 50 µM FFA (middle traces) or in the presence of 10 µM Ril + 50 µM FFA (bottom traces). The left and right panels in A illustrate two different plateau bursters under different indicated pharmacological treatments. Rhythmic burst activity of the plateau pacemaker was mostly affected (i.e., reduced) by blockade of $I_{CAN}$ channels with FFA (**A**), whereas the oscillatory pacemaker activity was more sensitive to persistent sodium channel ($I_{NaP}$) blockade with Ril (**B**). Correspondingly, longer duration plateau driven discharge in the mixed pacemaker phenotype was blocked with FFA alone, whereas remaining short-duration oscillatory bursts were blocked by the addition of Ril (**C**). Shaded activity at right of each trace: recording excerpts on expanded time-scale to facilitate comparison.

the model pacemaker neuron switched from inactive to plateauing states, then transcended a mixed plateau/oscillatory state until eventually the transition to a regularized oscillatory bursting condition occurred (*Figure 6E*). These in silico findings thus support the possibility that differences in proportion of the two conductances $I_{NaP}$ and $I_{CAN}$ could underlie the different discharge patterns expressed by actual preBötC pacemaker neurons. Additionally, they suggests that the developmental process responsible for the difference in functional composition of the inspiratory pacemaker population at

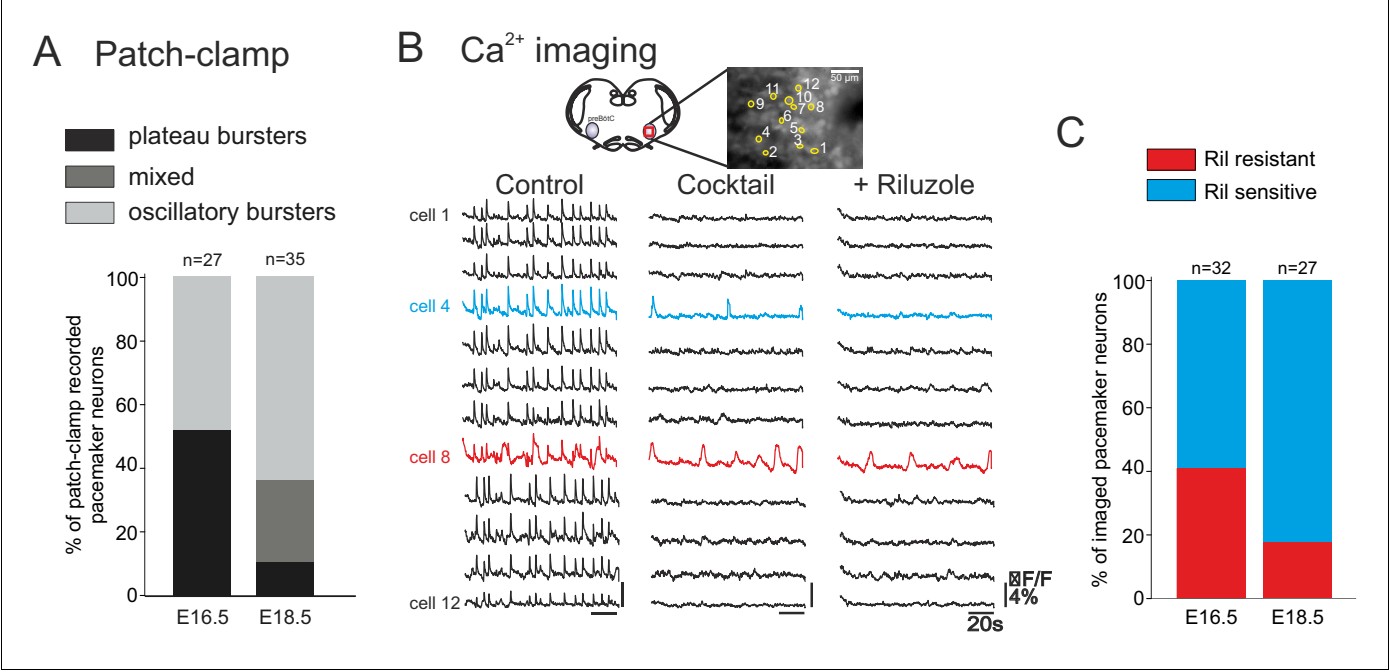

**Figure 5.** Developmental changes in inspiratory network pacemaker composition. (**A**) Bar histograms indicating the mean proportions (as% ) of individual pacemaker neuron subtypes recorded with patch-clamp at E16.5 (n = 27) and E18.5 (n = 35). Plateau bursting neurons, black shading; mixed plateau/oscillatory bursters, dark gray; oscillatory bursting neurons, light gray. (**B**) Calcium imaging protocol (see also *Figure 1*) to test the sensitivity of the preBötC pacemaker neuron subpopulation to $I_{NaP}$ blockade at E16.5 and E18.5. Calcium transients in 12 monitored inspiratory neurons at E16.5 in control conditions (left column), after synapse blockade (middle column), then under additional exposure to 10 µM riluzole (right column). Black traces correspond to non-pacemaker neurons (i.e., inactive under cocktail); the blue trace denotes a pacemaker neuron whose spontaneous activity was subsequently blocked under riluzole (i.e., an $I_{NaP}$-dependent pacemaker); the red trace corresponds to a pacemaker neuron that remained rhythmically activity under both cocktail and riluzole (i.e., an $I_{CAN}$-dependent pacemaker). (**C**) Bar histograms representing the proportions of riluzole-sensitive (blue bar) *vs* riluzole-resistant (red bars) neurons detected at E16.5 and E18.5. Note the higher proportion of riluzole-sensitive ($I_{NaP}$-dependent) pacemaker neurons at the later embryonic stage.

E16.5 and E18.5 could involve a switch in this proportion, and thereby resultant rhythmic burst patterning, within individual neurons.

## Changing role of pacemaker neurons in embryonic network rhythmogenesis

The change in pacemaker neuron conductance proportions during late embryonic development led us to also ask whether this transition is associated with age-dependent changes in the overall mechanism by which preBötC circuitry generates rhythmic output. To assess this possibility, we examined how overall network rhythm generation is affected by blockade of either $I_{NaP}$ or $I_{CAN}$, or both, at the two studied embryonic ages. Using transverse brainstem slice preparations, we recorded network electrical activity in control conditions and in the presence of Ril or/and FFA. In these experiments, where entire network activity was monitored, we used riluzole at 20 µM to ensure blockade of $I_{NaP}$ throughout the network. It should be noted, however, that very similar results were obtained with Ril applied at 10 µM. At E16.5, application of 50 µM FFA (*Figure 7A*, left panel) or 20 µM Ril (*Figure 7A*, right panel) significantly decreased the frequency of the ongoing spontaneous inspiratory rhythm by 46 ± 9% (*t*-test, p= 0.006) and 89.7 ± 5% (Mann-Whitney test, p<0.001), respectively (*Figure 7B*, left histogram bars). Unexpectedly, however, the same drug treatments performed at E18.5 had no significant effect (*t*-test, p= 0.1 and 0.4 for FFA and Ril treatments, respectively), with the frequencies of the ongoing preBötC rhythms in each case remaining unchanged under either FFA (*Figure 7C*, left middle trace) or Ril (*Figure 7C*, right middle trace) application (*Figure 7D*, left bars). Furthermore, when the drugs were applied concomitantly, rhythmic preBötC activity was fully

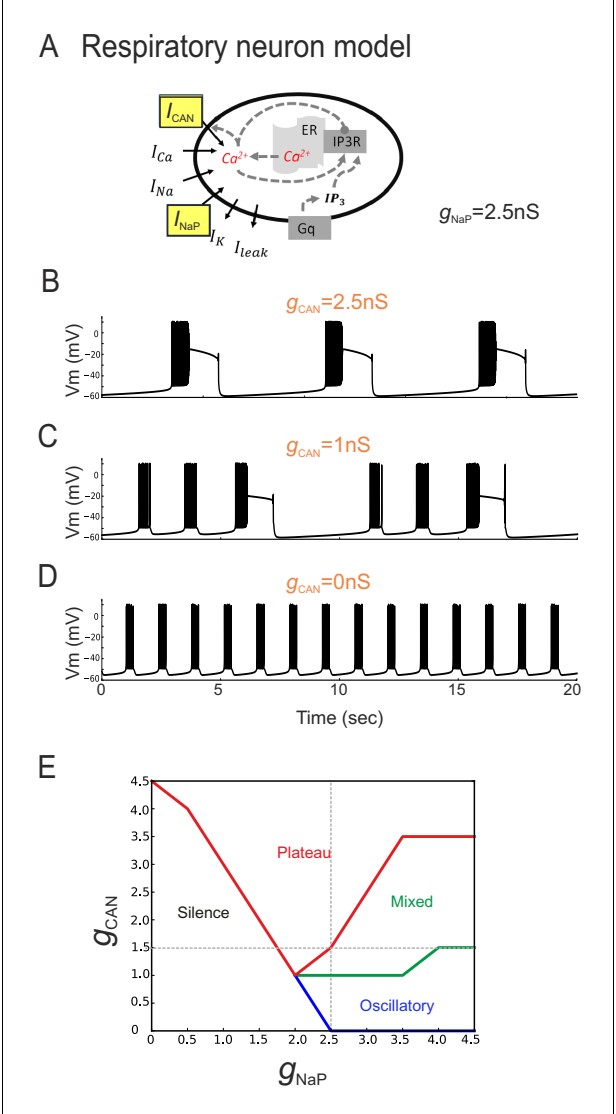

**Figure 6.** The balance between simulated $I_{CAN}$ and $I_{NaP}$ in silico can determine pacemaker discharge phenotype. (**A**) Computational model of an inspiratory pacemaker neuron with the main burst-generating currents $I_{CAN}$ and $I_{NaP}$ highlighted in yellow. For other abbreviations, see Material and methods and (*Toporikova et al., 2015*). (**B–D**) Model neuron voltage traces obtained with $I_{NaP}$ conductance ($g_{NaP}$) held constant at 2.5 nS and $g_{CAN}$ set to different steady state values (indicated in orange). Depending on $g_{CAN}$ magnitude, the model neuron generated plateau bursting discharge (B, $g_{CAN}$ = 2.5 nS), a mixed firing pattern (C, $g_{CAN}$ = 1 nS) or a rhythmic oscillatory burst pattern (D, $g_{CAN}$ = 0 nS). (**E**) Graph showing the type of model neuron discharge as a function of the proportions of $g_{NaP}$ and $g_{CAN}$ magnitudes. The gray dashed lines correspond to the values of constant $g_{NaP}$ and $g_{CAN}$ used, respectively, in the variable $g_{CAN}$ simulations illustrated in (**B-D**) and in complementary simulations in which $I_{NaP}$ was varied with $g_{CAN}$ held constant (data not shown). Depending on the balance between $g_{NaP}$ and $g_{CAN}$ amplitudes, the model neuron could exhibit the three different patterns observed with patch-clamp recordings in vitro.

blocked at E16.5 (*Figure 7A*, lower traces; 7B, right histogram bars; Mann-Whitney test, p<0.001), but rhythmic activity persisted at E18.5 (*Figure 7C*, lower traces) albeit at a significantly reduced cycle frequency (*Figure 7D*, right bars; Mann-Whitney test, p<0.01). Finally, to confirm that the effects involving FFA were not due to target actions other than on $I_{CAN}$ (*Guinamard et al., 2013*), we performed a series of equivalent experiments using 10 µM 9-phenanthrol instead of 50 µM FFA: comparable results at E16.5 (i.e., a frequency decrease (by 39 ± 15%; t-test, p= 0.05) when applied

alone and full blockade of rhythmic activity when co-applied with Riluzole; n = 3) and E18.5 (no significant effect on frequency when applied alone (t-test, p= 0.1) and persistence of rhythmic activity but at a lower frequency (63 ± 2%; t-test, p<0.001) when co-applied with Riluzole; n = 6) were obtained (data not shown). To ensure that the apparent reduced sensitivity of the network rhythm at E18.5 to blockers was not due to decrease in their diffusion into the older tissue, we repeated these experiments using 300 μm instead of 450 μm slices. However, the finding that the expression of rhythmic activity persisted in such thinner preparations in the presence of FFA + Ril, albeit at a lower frequency compared to control conditions (51.2 ± 15%; n = 6; Mann-Whitney test, p = 0.02) argues against this possibility. Altogether, these data therefore show that at E16.5, inspiratory network rhythmogenesis requires co-activation of $I_{NaP}$ and $I_{CAN}$, whereas later at E18.5, blockade of either conductance fails to fully prevent the generation of rhythmic activity. This in turn indicates that a developmental change has indeed occurred in the mechanistic basis for respiratory network rhythm generation, which evidently is linked to a less predominant role played by endogenous pacemaker neurons at the older embryonic stage.

To further explore this hypothesis, we sought evidence for a possible developmental increase in the role played by synaptic transmission in network rhythmogenesis by measuring the amplitude of the glutamatergic synaptic drive for inspiratory burst production in individual neurons recorded from slices at E16.5 and E18.5. Inspiratory cells were first identified in current clamp conditions when their spontaneous burst discharges occurred in phase with overall preBötC network activity. Cells were then held at −50 mV in voltage clamp and the synaptic currents generated during up to 15 consecutive inspiratory bursts were monitored and measured (*Figure 8A,B*, upper traces). The synaptic current magnitudes obtained at E16.5 and E18.5 (red traces in lower *Figure 8A and B*, respectively) were found to be significantly different (Mann-Whitney test, p<0.001), with the mean amplitude value obtained for 17 neurons at E18.5 (55 ± 0.9 nA) being >3 fold greater than that measured in 14 cells at E16.5 (15 ± 0.9 nA), (*Figure 8C*). The finding that excitatory post-synaptic currents do indeed appear to increase developmentally supports the hypothesis that the embryonic inspiratory network operates through different age-dependent strategies: one based principally on a subset of neuronal pacemakers and another in which overall circuit connectivity also plays a critical role.

## Discussion

Three main findings are reported in this study. First, neurons with endogenous pacemaker properties are present in the mouse embryonic preBötzinger network, constituting a functionally heterogeneous subpopulation in terms of their burst discharge patterns and underlying burst-promoting conductances $I_{NaP}$- and $I_{CAN}$. Second, these different pacemaker phenotypes are not invariably represented within the embryonic network during the prenatal period, but their relative proportions change as prenatal development progresses. Third, the overall mechanism responsible for preBötC network rhythmogenesis also changes during embryonic development, switching from a purely pacemaker neuron-driven, $I_{NaP}/I_{CAN}$-dependent mechanism to an emergent network process in which both neuronal pacemakers and circuit excitatory synaptic connectivity appear to play important roles. Presumably, the establishment of these two mechanisms acting in cooperation serves to ensure that a fully effective respiratory motor command is available at the time of birth.

### Presence of pacemaker neurons in the embryonic respiratory network

Here, we describe for the first time distinct pacemaker properties expressed by a subpopulation of inspiratory neurons in the embryonic preBötC network. As also found in early postnatal rodents (*Thoby-Brisson and Ramirez, 2001*; *Del Negro et al., 2002*; *Pena et al., 2004*; *Del Negro et al., 2005*), two major types of prenatal preBötC pacemaker neurons are present: one whose bursting activity depends mainly on the activation of a Ril-sensitive, voltage-dependent persistent sodium current ($I_{NaP}$) and another that is dominated by a FFA-sensitive, $Ca^{2+}$-activated nonspecific cationic current ($I_{CAN}$). Furthermore, a proportion of pacemaker neurons express bursting activity that arises from both mechanisms, indicative of an intermediary pacemaker phenotype. We also found that in the embryonic network, the two main pacemaker mechanisms produce a depolarizing potential drive with distinctly different durations and cycle frequencies - $I_{CAN}$ is prevalent in generating long and slowly repeating plateau-like activity, while $I_{NaP}$ tends to underlie shorter and more rapid burst discharge - thus indicating that the pacemaker subtypes are distinguishable according to both the

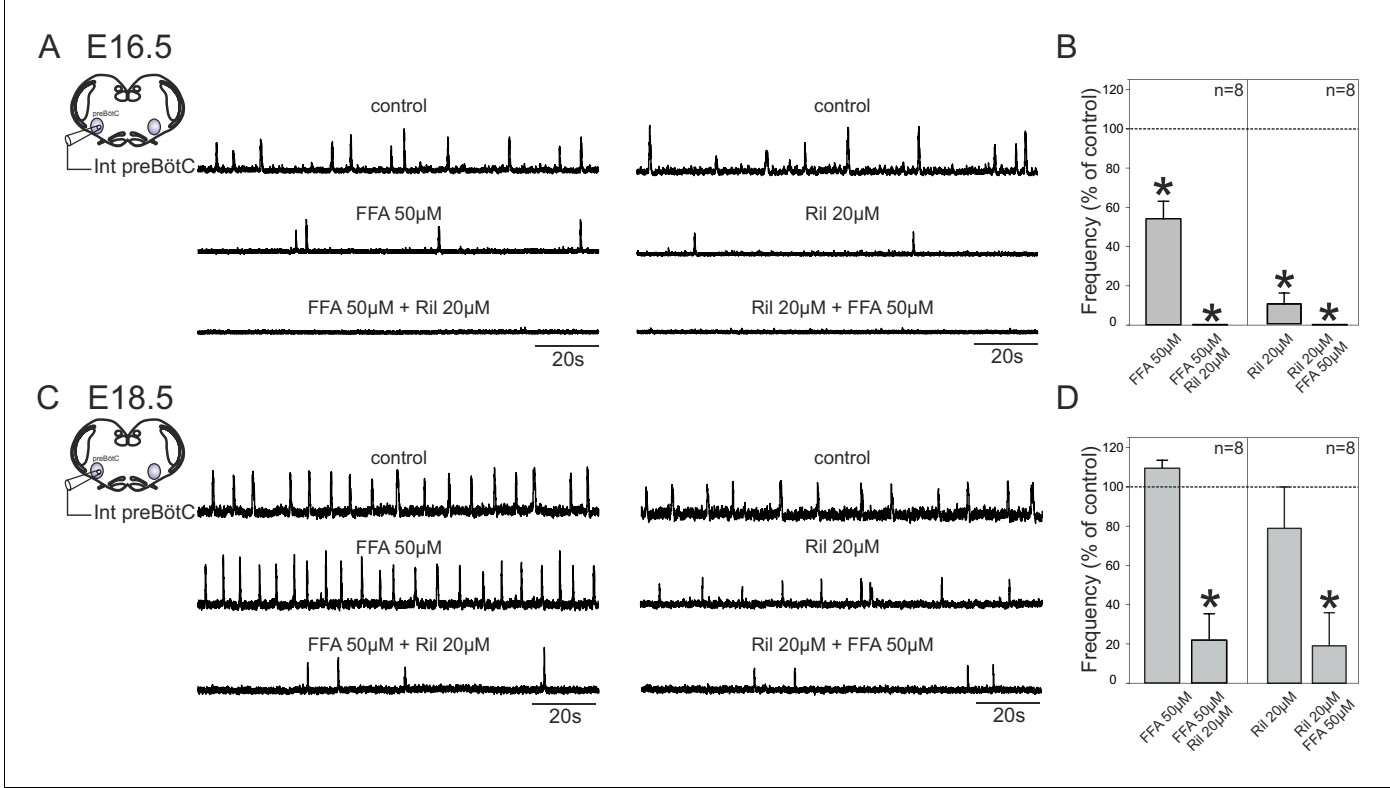

**Figure 7.** Mechanisms for inspiratory network rhythmogenesis change during embryonic development. (A) Schematic representation of an in vitro slice preparation used for electrophysiological recording of spontaneous preBötC network activity (integrated traces, Int preBötC) in control conditions (top traces) or in the presence of either FFA (50 µM, middle trace, left) or Ril (20 µM, middle trace, right) alone or in combination (bottom traces) at E16.5. (B) Bar histograms representing the percentages of cycle frequency change (mean ± SEM) compared to control under the different pharmacological conditions illustrated in A. Asterisks indicate significant differences (t-test, $p<0.05$); n = number of experiments in each case. (C,D) same layout as (A,B) for experiments performed at E18.5. Note that unlike at E16.5 where both FFA and Ril significantly reduced rhythmogenesis, blockade of either $I_{CAN}$ or $I_{NaP}$ alone did not significantly perturb rhythm generation at E18.5. Also, blockade of both conductances at E16.5 was sufficient to completely suppress rhythm generation, but not at E18.5.

conductance mechanisms responsible for the initiation of their bursts as well as for burst termination. Burst terminating mechanisms have not yet been fully identified in neonatal inspiratory network neurons, but it has been suggested that Na- and ATP-dependent outward currents play a key role (*Krey et al., 2010*) together with $I_{NaP}$ and $I_{CAN}$ inactivation/deactivation processes (*Del Negro et al., 2002*; *Del Negro et al., 2005*) and the activation of voltage- and calcium-dependent K+ conductances (*Butera et al., 1999*). Additionally, we observed that plateau-like inspiratory pacemaker bursting is considerably more prevalent at earlier embryonic ages (representing ~50% of the total pacemaker population) but is then largely superseded by faster, short-duration oscillatory bursting as development progresses. Perhaps unsurprisingly, the shape, duration and regularity of inspiratory pacemaker bursts occurring at E18.5, and in contrast to the considerably slower, yet robust, plateauing activity expressed at E16.5, more closely resemble the discharge patterns expressed by preBötC pacemakers in the neonatal animal (cf., *Pena et al., 2004*; *Del Negro et al., 2005*). It is likely, therefore, that together with developmental changes in the balance of membrane conductances engaged in initiating burst-driving potentials (see below), the conductance mechanisms required for their appropriate cycle-by-cycle termination in the postnatal respiratory network are also being established and refined at late embryonic stages.

## Developmental plasticity in pacemaker neuron heterogeneity

Our cell patch clamp and imaging data associated with differential pharmacological manipulation revealed that both $I_{NaP}$- and $I_{CAN}$-dependent pacemaker neuron subtypes are equally present at

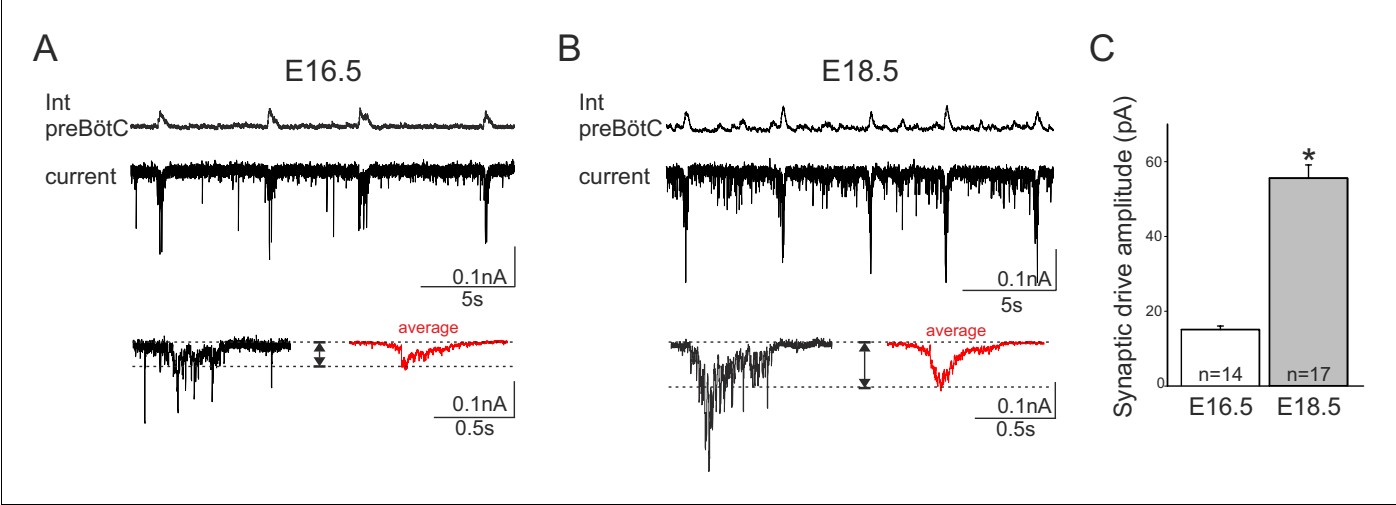

**Figure 8.** Developmental increase in burst-generating synaptic excitation amongst inspiratory network neurons. Integrated preBötC network activity (top traces, Int preBötC) with simultaneous intracellular recordings of membrane current fluctuations (lower traces) in rhythmically active preBötC neurons under voltage clamp (holding potential set at −50 mV) obtained at E16.5 (**A**) and E18.5 (**B**). Expanded bottom traces in A and B show the excitatory synaptic currents (left side in each case, single burst cycle; right side in red, average of 15 cycles) recorded from individual inspiratory neurons during rhythmic network activity. Dashed lines indicate baseline and maximum amplitude levels of the synaptic currents. Note their larger amplitude at E18.5 for both single cycle and averaged events. (**C**) Bar histograms representing the mean maximal amplitude (± SEM) of excitatory synaptic current underlying burst discharge measured in 14 neurons at E16.5 (unshaded bar) and 17 neurons at E18.5 (gray bar). The asterisk indicates significant difference (p< 0.001).

E16.5. However, the proportion of $I_{NaP}$-dependent (Ril-sensitive) pacemakers increases significantly by E18.5, with FFA-sensitive or Ril-resistant (therefore presumed $I_{CAN}$-dependent) cells being rarely discernible at this later stage. Significantly, previous studies have reported that cadmium-sensitive (therefore presumed $I_{CAN}$-dependent) pacemaker neurons are more readily detected in the preBötC network of juvenile mice (P8 - P10) than at neonatal (P0 – P5) stages (*Pena et al., 2004*) where they were found to represent < 1% of recorded inspiratory cells compared to 7.5% in the older animals (*Del Negro et al., 2005*). These observations are therefore consistent with our own finding that in a developmental period around the time of birth embryonic $I_{CAN}$-dependent inspiratory pacemakers are much less prevalent than $I_{NaP}$-dependent ones. However, the functional significance of the re-emergence of the $I_{CAN}$-dependent pacemaker subtype during the second post-natal week (*Pena et al., 2004*; *Del Negro et al., 2005*) remains unknown.

The possibility that the developmental changes in the discharge patterns of individual inspiratory pacemaker neurons could be a direct consequence of changes in the balance between $I_{NaP}$ and $I_{CAN}$ expression was supported by our simulation data. Increasing the strength of membrane $g_{NaP}$ relative to $g_{CAN}$ (or *vice versa*) within a single model pacemaker neuron could reversibly switch its spontaneous rhythmogenic activity from slow, plateau-like drive potentials as found at E16.5 to the accelerated, short-duration oscillatory bursting observed at E18.5. Whether an equivalent developmental alteration in channel balance from an $I_{CAN}$ to $I_{NaP}$ predominance occurs at the single cell level in real embryonic pacemaker neurons, or that $I_{CAN}$-dependent pacemakers within inspiratory network are replaced by a de novo population of $I_{NaP}$-dependent pacemakers, remains unknown. Although such processes are not mutually exclusive, the finding that a small group of our in vitro recorded pacemaker cells could express a mixture of both plateau- and oscillatory-based discharge patterns, as similarly produced by our model neuron with intermediate $g_{NaP}$ and $g_{CAN}$ proportions (see also *Dunmyre et al., 2011*), was indicative of a mechanistic switch that was transitioning within individual neurons.

## Pacemaker neuron- vs network-driven rhythmogenesis

What role does the embryonic pacemaker neuron subpopulation play in respiratory network rhythmogenesis? Two observations in the present study are relevant to this question. First,

**Table 1.** Parameters for single cell model.

| Parameter related to: | Parameter values: |
| --- | --- |
| $I_{NaP}$ | $g_\mathrm{NaP} = 0 - 3\,\mathrm{nS}$; $v_h = -48$ mV; $s_h = 5$ mV; $\overline{\tau_h} = 10000$ ms; $V_\mathrm{NaP} = 50$ mV; |
| $I_{Na}$ | $g_\mathrm{Na} = 28$ nS, $v_\mathrm{Na} = 50$ mV; $v_m = -40$ mV; $s_m = -6$ mV; $\overline{\tau_n} = 10$ ms |
| $I_{CAN}$ | $g_\mathrm{can} = 0\ to\ 4.5$ nS; $K_\mathrm{can} = 0.74$ µM |
| $I_K$ | $g_\mathrm{K} = 11.2$ nS; $V_\mathrm{K} = -65$ mV |
| $I_L$ | $g_L = 2.7$ nS; $V_L = -60$ mV |
| $I_{Ca}$ | $g_\mathrm{ca} = 0.05$ nS; $V_\mathrm{ca} = 150$ mV; $\alpha = 0.055$; $V_\mathrm{PMCA} = 2$; $K_\mathrm{PMCA} = 0.3$ |
| ER Ca | $\lambda = 0.03$; $f_i = 0.0001$; $[\mathrm{IP_3}] = 1$ µM; $V_i = 4$ µM$^3$ <br> $A = 0.0005$; $V_\mathrm{serca} = 400$ aMol/s; $K_\mathrm{serca} = 0.2$; $\sigma = 0.185$; $L_\mathrm{IP3R} = 0.37\,\mathrm{pL/s}$ ; $P_\mathrm{IP3R} = 31000$ pL/s; <br> $K_I = 1.0$ µM; $K_d = 0.4$ µM; $K_a = 0.4$ µM |
| Other | $Cm = 21\,\mathrm{pF}$ |

pharmacological attenuation of $I_\mathrm{CAN}$ and $I_\mathrm{NaP}$ at E16.5 completely blocked rhythmic preBötC network activity, whereas the same treatment of slices at E18.5 reduced, but failed to completely suppress ongoing network activity. Notwithstanding the fact that the vast majority of inspiratory network neurons, including pacemakers and non-pacemakers alike, possess both NaP and CAN conductances (*Del Negro et al., 2011*) and are therefore affected by our drug treatments, these findings indicate that a fundamental developmental change had occurred whereby network rhythmogenesis has shifted from an obligatory pacemaker-driven mechanism to one in which pacemaker neurons were no longer uniquely essential. Second, as shown by in vitro experimental data, the strength of excitatory glutamatergic synaptic transmission is significantly stronger throughout the preBötC circuitry of E18.5 compared to E16.5 slices. On this basis, therefore, we propose that at earlier embryonic stages, pacemaker neurons play an essential role in network rhythm generation but that later in the prenatal period, the contribution of glutamatergic synaptic interactions becomes increasingly important so that nearer the time of birth, both cellular pacemaker properties and synaptic excitation are critical to network rhythmogenesis. Interestingly, such a developmental plasticity occurring in the late embryo embodies essential features of both the 'pacemaker neuron' (*Pena et al., 2004*; *Ramirez et al., 2011*) and 'group-pacemaker' hypotheses (*Rekling and Feldman, 1998*; *Pena et al., 2004*; *Feldman and Del Negro, 2006*) that have been proposed to account for respiratory rhythm generation in the early postnatal rodent (for review, see *Feldman et al., 2013*).

In conclusion, the present study provides the first description of a heterogeneous population of neurons bestowed with pacemaker properties in the embryonic respiratory network. The composition of this population, both in terms of pacemaker subtypes and their implication in overall network rhythmogenesis undergo significant maturational changes during the prenatal period. As yet, we have not been able to establish a direct link between these two transitional processes, nor has our data allowed us to determine the specific contribution that each pacemaker subtype makes to inspiratory network output as a whole. However, our in vitro results from differential pharmacology lead to the conclusion that the pacemaker cell subpopulation is exclusively responsible for network rhythmogenesis at earlier embryonic stages, but then assumes a cooperative role with recurrent synaptic excitation throughout the wider network immediately prior to birth. A better understanding of such prenatal developmental processes could in turn provide relevant insights into clinical aspects of early post-natal respiratory disorders.

## Materials and methods

All animal procedures were conducted in accordance with the local ethic rules of the University of Bordeaux as well as national and European committee regulations. Experiments were performed on mouse embryos of either sex obtained from pregnant OF1 females raised in our laboratory's breeding facility.

## Rhythmic slice preparations

Brainstem transverse slice preparations that isolated the preBötC network were obtained from mouse embryos using the following procedures: pregnant mice were killed by cervical dislocation on embryonic day (E) 16.5 or E18.5, the day of the plug being considered as E0.5. Embryos were excised from the uterine horns and their individual uterine bags, and prior to experimental use, were either placed in artificial cerebrospinal fluid (aCSF, see below) that was continuously supplemented with oxygen at a temperature not exceeding 24°C for E16.5 embryos or were manually stimulated to activate spontaneous breathing behavior and then kept under a warm heating light for E18.5 embryos. Slice preparations were dissected in cold oxygenated aCSF composed of (in mM): 120 NaCl, 8 KCl, 1.26 $CaCl_2$, 1.5 $MgCl_2$, 21 $NaHCO_3$, 0.58 $NaH_2PO_4$, 30 glucose, pH 7.4. In a first step, the hindbrain was isolated from the embryo's body by a rostral section made at the level of the rhombencephalon and a caudal section made at the level of the first cervical roots. The isolated hindbrain was then placed in a low melting point agar block and carefully oriented to enable serial transverse sectioning in a rostral-to-caudal direction using a vibratome (Leica, Germany). A 450-μm-thick slice, with its anterior limit set 250–300 μm caudal to the more caudal extension of the facial nucleus, was isolated. Other anatomical landmarks, such as the wide opening of the fourth ventricule, the presence of the inferior olive, the nucleus ambiguus and the hypoglossal nucleus were also used to determine the appropriate sectioning axis (as also referred to in newborn mice by *Ruangkittisakul et al., 2011*). Such slice preparations encompass the region containing a significant portion of the preBötC network capable of spontaneously generating fictive rhythmic inspiratory activity (*Thoby-Brisson et al., 2005*; *Toporikova et al., 2015*) are devoid of the more rostral Bötzinger complex and contain only the most rostral region of the more caudal rVRG. On this basis, therefore, all neuronal populations examined were considered to be integral components of the pre-Bötzinger network. Slices were then transferred to a recording chamber, rostral surface upwards, and continuously superfused with oxygenated aCSF at a temperature of 30°C. Preparations were allowed to recover from the slicing procedure and subsequent calcium indicator loading (see below) for 20–30 min before any recording sessions commenced.

## Recording procedures

Global preBötzinger network activity in slice preparations was recorded using glass micropipettes (tip diameter 80–100 μm) positioned on the upper surface of the slice in a region ventral to the nucleus ambiguus where respiratory circuitry is located. The micropipettes were fabricated from aCSF-filled borosilicate glass tubes (Harvard Apparatus, Germany) broken at the tip and used as suction electrodes connected to a high-gain amplifier (AM Systems, USA). The collected signals were filtered (bandwidth 3 Hz - 3 kHz), integrated (time constant 100 ms; Neurolog, Digitimer, England) and stored on a computer via a Digidata 1440 interface and PClamp10 software (Molecular Devices, USA). The stored files were analyzed off-line.

Whole cell patch-clamp recordings of pacemaker neurons were performed under visual control using differential interference contrast. Patch pipettes were fabricated with borosilicate glass tubes using a puller (Sutter Instrument, USA) and filled with a solution composed of (in mM): 140 K-gluconate acid, 1 $CaCl_2.6H_2O$, 10 EGTA, 2 $MgCl_2$, 4 $Na_2ATP$, 10 HEPES, pH 7.2 and had tip resistance of 5–7 MΩ when filled with this solution. Electrophysiological signals were recorded using an Axoclamp 2A amplifier (Molecular Devices) and the same digitizing interface and software as stipulated above. Neurons were selected for patch-clamp recording on the basis of their rhythmic impulse discharge that had to be in phase with the extracellularly-recorded population activity both in control conditions and under conditions where network chemical synapses were subsequently blocked (by a cocktail of pharmacological agents, see below). Recorded neurons were located close to the extracellular macroelectrode or in the contralateral preBötC network that had been initially identified in control conditions as exhibiting rhythmically organized fluorescent changes in phase with the electrophysiological recording of its contralateral network partner. Only recordings lasting longer than 8 min were considered for the burst parameter analysis presented in *Figure 2*. I-V curves were plotted by measuring the amplitude of the membrane current evoked by 0.5 s duration voltage steps from −100 to +30 mV. Current amplitude was measured at steady state, 400 ms after the onset of a given voltage step. The magnitude of the synaptic drive underlying individual inspiratory bursts that results from the summation of glutamatergic excitatory post-synaptic currents was measured as the maximal

amplitude of the envelop (i.e., the difference between the horizontal dashed line pairs in *Figure 8*) and averaged for 15 consecutive bursts per neurons in slices at E16.5 and at E18.5.

## Calcium imaging

The method used for calcium imaging that allows monitoring the activity of multiple preBötC neurons simultaneously has been fully described elsewhere (*Thoby-Brisson et al., 2005*; *Toporikova et al., 2015*). Briefly, slice preparations were first incubated in the dark for 45 min at room temperature in a solution of oxygenated aCSF containing the cell-permeable calcium indicator dye Calcium Green-1 AM (10 µM; Life Technologies, France). After dye loading, preparations were positioned rostral side up in the recording chamber. Before subsequent image acquisition, a 30 min delay was observed to wash out excess dye and enable the preparation to stabilize in oxygenated aCSF at 30°C. Fluorescent signals were captured through a FN1 upright microscope (Nikon, Japan) equipped with an epifluorescent illumination system and a fluorescein filter coupled to an Exiblue camera (QImaging, Surrey, Canada). Images (100 ms exposure time) were acquired over periods lasting 120 s and analyzed using customized software provided by Dr N. Mellen (*Mellen and Tuong, 2009*).

## Pharmacology

Pharmacological agents were obtained from Sigma or Tocris (France) and dissolved in oxygenated aCSF for bath-application for 15 – 30 min at their final concentration of 10–20 µM Riluzole (Ril) to block persistent sodium currents ($I_{NaP}$), and 50 µM Flufenamic acid (FFA) to block calcium-dependent non-specific cationic currents ($I_{CAN}$). Although such a relatively low FFA concentration was used in order to limit potential non-specific effects (*Guinamard et al., 2013*), complementary experiments were also performed in which CAN conductances were blocked by 10 µM 9-phenanthrol in place of FFA. The effects of all drugs on rhythm generation were assessed at the end of each exposure period. To synaptically isolate respiratory network neurons, we applied a cocktail of synaptic blockers containing 20 µM 6-cyano-7-nitroquinoxaline-2,3-dione (CNQX), 10 µM DL-2-amino-5-phosphonovaleric acid (AP5), 1 µM strychnine and 10 µM bicuculline. Burst frequency and the amplitude and duration of the depolarizing drive potentials underlying bursts are given as mean ± SEM. Statistical significance was assessed by Student's *t*-test, or Mann-Whitney. Mean values were considered as statistically different when p<0.05. Differences in pacemaker type proportions were assessed using the Chi-square test and considered significant for p<0.05.

## Modeling procedures

### Single cell model

The single cell simulation of an embryonic inspiratory neuron is based on a previously published model (*Toporikova et al., 2015*). Briefly, membrane voltage (V) is determined using a balance of essential burst generating currents (*Toporikova and Butera, 2011*): a voltage-gated persistent Na$^+$ current ($I_{NaP}$) and a K$^+$-dominated passive leakcurrent ($I_K$), a non-inactivating voltage-gated calcium current ($I_{Ca}$), and a calcium-activated nonspecific cation current ($I_{CAN}$). Accordingly, the voltage component of the model is described by the following equation:

$$C_m \frac{dV}{dt} = -I_{NaP} - I_K - I_{Ca} - I_{CAN} - I_h \tag{1}$$

where the individual currents are given by:

$$I_{NaP} = g_{NaP} m_\infty h (V - V_{NaP}) \tag{2}$$

$$I_{Ca} = g_{Ca} m_\infty (V - V_{Ca}) \tag{3}$$

$$I_{CAN} = g_{CAN} \frac{[Ca]}{[Ca] + K_{CAN}} (V - V_{NaP}) \tag{4}$$

$$I_K = g_K (V - V_K) \tag{5}$$

$$I_{\mathrm{h}} = g_{\mathrm{h}} n_\infty (V - V_{\mathrm{h}}) \tag{6}$$

Here, the parameter $g$ represents the maximal conductance of each current (Na, NaP, K, CAN, or H), while m and n are activation variables and h is the inactivation variable. To reduce the number of parameters in the model, the activation of $I_{\mathrm{Ca}}$ was approximated with the activation function for $I_{\mathrm{NaP}}$. This fast, non-inactivated current is intended to track average changes in $\mathrm{Ca}^{2+}$ during a burst.

The inactivation variable h is described by the following equation:

$$\frac{dh}{dt} = \frac{h_\infty(V) - h}{\tau_h} \tag{7}$$

where $h_\infty(\mathrm{V})$ defines the steady state activation/inactivation curve and $\tau_{\mathrm{h}}$ is the voltage-dependent time constant. The steady state activation/inactivation curves and synaptic activation function H(V) are modeled as a sigmoid, with x representing m, n, s and h:

$$x_\infty = \frac{1}{1 + \exp\left(\frac{V - V_x}{s_x}\right)} \tag{8}$$

while the time constant is modeled as follows

$$\tau_x = \frac{\overline{\tau_x}}{\cosh\left(\frac{V - V_x}{2 s_x}\right)} \tag{9}$$

We used a two-pool model to account for $\mathrm{Ca}^{2+}$ fluxes through the cell's plasma membrane and endoplasmic reticulum (ER). Therefore, $\mathrm{Ca}^{2+}$ kinetics are modeled by three equations representing the intracellular $\mathrm{Ca}^{2+}$ ($[\mathrm{Ca}]_i$) balance, ER the $\mathrm{Ca}^{2+}$ ($[\mathrm{Ca}]_{\mathrm{ER}}$) balance and IP$_3$ the channel gating variable (l) (*Li and Rinzel, 1994*).

$$\frac{d[\mathrm{Ca}]_i}{dt} = \frac{f_i}{V_i}\left(\frac{1}{\lambda}\left(J_{PM_{IN}} - J_{PM_{OUT}}\right) - \left(J_{ER_{IN}} - J_{ER_{OUT}}\right)\right) \tag{10}$$

$$\frac{d[Ca]_{TOT}}{dt} = \frac{f_i}{V_i}\left(\frac{1}{\lambda}\left(J_{PM_{IN}} - J_{PM_{OUT}}\right)\right) \tag{11}$$

$$\frac{dl}{dt} = A\left(K_d - l\left([\mathrm{Ca}]_i + K_d\right)\right) \tag{12}$$

where $\lambda$ is the ratio of ER to plasma membrane surfaces, $\lambda = \frac{A_{ER}}{A_{pm}}$, $f_i$ is a constant reflecting the ratio of bound-to-free $\mathrm{Ca}^{2+}$ concentration normalized to effective area (*Wagner and Keizer, 1994*), $A$ is a scaling constant, $K_d$ is the dissociation constant for IP$_3$ receptor inactivation by $\mathrm{Ca}^{2+}$. The ER calcium concentration is calculated as $[\mathrm{Ca}]_{\mathrm{ER}} = \frac{[Ca]_{TOT} - [\mathrm{Ca}]_i}{\sigma}$, where $\sigma$ is the ratio of cytosolic to ER volumes.

The flux into the cytosol from the endoplasmic reticulum (ER) ($J_{ER_{IN}}$) is regulated by the activity of IP$_3$ receptors and is defined as:

$$J_{ER_{IN}} = \left(L_{IP_3R} + P_{IP_3R}\left[\frac{[IP_3][Ca]_i\, l}{([IP_3] + K_I)([Ca]_i + K_a)}\right]^3\right)\left([Ca]_{ER} - [Ca]_i\right) \tag{13}$$

where $P_{\mathrm{IP3R}}$ is the maximum total permeability of IP$_3$ channels, $L_{\mathrm{IP3R}}$ is the ER leak permeability, $[\mathrm{IP}_3]$ is the IP$_3$ concentration, $K_I$ and $K_a$ are the dissociation constants for IP$_3$ receptor activation by IP$_3$ and $\mathrm{Ca}^{2+}$, respectively.

The $\mathrm{Ca}^{2+}$ flux from the cytosol back to the ER ($J_{ER_{OUT}}$) is controlled by the activity of SERCA pumps:

$$J_{ER_{OUT}} = V_{SERCA} \frac{[Ca]_i^2}{K_{SERCA}^2 + [Ca]_i^2} \qquad (14)$$

where $V_{SERCA}$ is the maximal SERCA pump rate, $K_{SERCA}$ is the coefficient for the SERCA pumps.

The influx of Ca²⁺ through the plasma membrane is proportional to the voltage-gated Ca²⁺ current

$$j_{PM_{IN}} = -\alpha\, I_{ca} \qquad (15)$$

where $\alpha$ is a proportionality constant.

The outflux of Ca²⁺ through the plasma membrane is controlled by the activity of PMCA pumps as follows:

$$j_{PM_{OUT}} = V_{PMCA} \frac{c^2}{K_{PMCA}^2 + c^2} \qquad (16)$$

where $V_{PMCA}$ is the maximum activity of PMCA pumps and $K_{PMCA}$ is the kinetic constant for PMCA pumps.

The differential equations were solved numerically using *Python* programming language. Numerical integration was carried out using the *odeint* function of the *Scypy* library with an adaptable time step. This method solves standard differential equations using the *lsoda* method for stiff or non-stiff systems. To ensure that the operational single cell model had attained a stable oscillatory regime, the first 6 s of simulations were systematically discarded.

## Acknowledgements

This work was supported by the Institut National de la Santé et de la Recherche Médicale to MTB and the Agence Nationale de la Recherche (ANR12-BSV4-0011-01) to MTB. We are thankful to the members of the OASM research team for their valuable comments and helpful discussions throughout this study.

## Additional information

### Funding

| Funder | Grant reference number | Author |
|---|---|---|
| Agence Nationale pour le Développement de la Recherche en Santé | ANR12-BSV4-0011-01 | Muriel Thoby-Brisson |

The funders had no role in study design, data collection and interpretation, or the decision to submit the work for publication.

### Author contributions

MC, Performed research and analyzed data; NT, MT-B, Designed research, Performed research and analyzed data, Wrote the paper; JS, Designed research, Wrote the paper

### Author ORCIDs

Muriel Thoby-Brisson, http://orcid.org/0000-0003-3214-1724

### Ethics

Animal experimentation: All experiments were performed in accordance with the guidelines of the European and French National legislation on animal experimentation and the local ethics committee of the University of Bordeaux (permit number 5012031A).

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
