## [Decision Letter]

Thank you for submitting your article "Development of pacemaker properties and rhythmogenic mechanisms in the embryonic respiratory network" for consideration by *eLife*. Your article has been reviewed by three peer reviewers, one of whom, Ronald Calabrese, is a member of our Board of Reviewing Editors and the evaluation has been overseen by Eve Marder as the Senior Editor. The following individuals involved in review of your submission have agreed to reveal their identity: Gregory D Funk (Reviewer #2); Naohiro Koshiya (Reviewer #3).

The reviewers have discussed the reviews with one another and the Reviewing Editor has drafted this decision to help you prepare a revised submission.

Summary:

In this interesting study, the authors perform a pharmacological, electrophysiological and calcium imaging analysis of pacemaker neuron membrane properties in the inspiratory center of the brainstem [preBötzinger complex network (preBötC)] during late embryonic development in the mouse. This work is supported by computational modeling of individual pacemaker neurons and the preBötC network. They demonstrate the existence of heterogeneous pacemaker oscillatory properties relying on distinct combinations of burst-generating INaP and ICAN conductances. The respective proportion of the different inspiratory pacemaker subtypes changes during prenatal development. They provide evidence that network rhythmogenesis switches from a purely INaP/ICAN-dependent mechanism at E16.5 to a combined pacemaker/network driven process at E18.5. The results have important implication for the maturation of the respiratory inspiratory pacemaker network across mammals and may potential impact our understanding of early postnatal failures of respiratory drive. The approach is novel: extending studies to the embryo to understand the post-natal developmental trajectory. The data are important contributions and will be of wide interest. There are 4 major concerns, however, that must be addressed by the authors before publication in *eLife*.

Essential revisions:

1) The network model should be removed for the following reasons: i) The model is not needed to provide rationale for the experiments testing for an increased role for synaptic interactions of in rhythm generation. The data themselves suggest this possibility. ii) The model parameters are highly underspecified so it has little (if any) predictive value, nor does it enhance understanding of network organization; and, iii) It is distracting, requiring a lot of effort to understand the model rather than focusing on the relatively simple conclusion that is based on the data (i.e., rhythmic inspiratory synaptic drive to inspiratory neurons increases developmentally).

2) The authors should provide a clear description of how they classified their neurons into the three activity groups (long-lasting plateau-like oscillations with impulses occurring at the beginning of the plateau followed by a depolarization block, short-lasting oscillations and bursts, mixture of long- and short lasting bursts) and mapped these onto biophysical properties. More raw data should be provided of voltage recordings and voltage-clamp records. Details of biophysical measurements should be provided, in tabular form if necessary, on passive properties (input resistance and capacitance) and active currents (current dynamics). Necessary details of the experimental procedures were not always provided, e.g. when FFA and Ril were given sequentially (e.g., Results section, subsection “Membrane conductance underlying the different discharge patterns”: "Ril alone"), how was washout of the first drug assured so the second blocker's effect could be assessed independently? More detailed quantification and statistics are needed for the groupings. For example, when the authors say "bursting activity of 10 of these was completely blocked" we would like to know the definition and measurement method for a burst or its duration or existence. Moreover, claiming block may seem reasonable from the typical traces provided, but the conclusion remains subjective unless documented quantitatively. While the study deals with the neural system's maturation, embryonic ages of the preps were not always clear. Since this part is fundamental to the rest of study, it could be documented less anecdotally and more quantitatively and rigorously.

3) A significant aspect of the study is based on two sets of observations: 1) imaging and/or patch-clamp demonstrations of burster types and blockers' effects, and 2) blockers' variable effects on the preBötC rhythms recorded locally. These two observations are, however, independent of each other and the causality remains an open question. The cells that were imaged and/or patch-clamped were near the upper surface of the slice where the blockers had ready access. The vast majority of the rhythmogenic population might have been left unaffected, especially in the 450 µm thick slice that can capture a large rostrocaudal substrate in a fetus brainstem. Given, theoretically, 10% of the cross-connected population is sufficient to be pacemakers for the population to produce rhythm (Butera et al. 1999b), a relatively large population unaffected by the blocker may be sufficient to maintain the rhythm such as ones shown in Figure 7. Since preBötC inspiratory pacemaker neurons' contralaterally projecting axons take a rostrally arched course (Koizumi et al. 2013), those pacemakers near the rostral cut-end affected with the blockers might have had lost recurrent connections to a half of the rhythmogenic population (thus loss of their cellular rhythm could have less impact on the rhythm inside slice). The main impact of this diffusion issue in the context of this study is that one of major conclusions is the rhythm at E18.5 is no longer dependent on pacemaker properties. Given that diffusion of drugs through brain tissue decreases as it matures, it is possible that the persistence of rhythm at E18.5 reflects that Ril and FFA did not diffuse through the entire slice. These possibilities should be addressed forthrightly; either by new analysis, a clever but relatively quick set of experiments designed to show that diffusion of drugs is not a problem, or simply qualifying their conclusions with appropriate caveats in Discussion.

4) Over-interpretation of the data is significant concern. This concern can be addressed with revisions to the text, following the minor comments of the reviewers, which are appended. Importantly, in terms of overall significance to understanding mechanisms of rhythm generation it is important to remind readers that these data are based on analysis of the network isolated in vitro.

[Editors' note: further revisions were requested prior to acceptance, as described below.]

Thank you for resubmitting your work entitled "Development of pacemaker properties and rhythmogenic mechanisms in the mouse embryonic respiratory network" for further consideration at *eLife*. Your revised article has been favorably evaluated by Eve Marder (Senior Editor), a Reviewing Editor, and two reviewers.

The manuscript has been improved but there are some remaining issues that need to be addressed before acceptance, as outlined below:

1) In their response to the past review the authors state "[…]performed a new series of experiments in which FFA and Riluzole were applied to a thinner (300µm) E18.5 slice preparations." and they claim similar results to 450 µm slice. The authors should include these data in the text, and they should describe the effects of FFA and Riluzole on the thinner slices. These data should be introduced with the caveat about differential diffusion with age. E.g. 'To ensure that reduced sensitivity of rhythm at E18.5 to blockers was not due to reduced diffusion of blockers into the older tissue, we repeated these experiments in thinner slices[…]'

2) The authors should provide more clarification about how they classified the three types of cell activities they observed in Figure 2: burst, plateau, mixed. The problem with the current presentation is that there is some overlap in the ranges of Figure 2. The authors should consider plotting their data set of Figure 2 on a burst duration vs. burst amplitude coordinate system, using a different color for each preparation categorized as plateau or burst. This will show the clouds and the overlap and give the skeptical reader a clear idea what is going on. This can be a new figure.

The mixed cells are somewhat of a problem because average data cannot capture their properties. Qualitatively, the plateaus are easy to distinguish from the bursts; the data are not completely clean, but we see that there is no problem telling the bursts from the plateaus when they occur one without the other in different cells. The problem comes with the mixed cells. Here the verbal and quantitative descriptions and the cloud discrimination break down requested above. This is because average data cannot capture the bimodal nature of the distributions. The text is inadequate on this point because it reports averages and SD for the bimodal data. The authors should recognize explicitly the bimodal nature and try to separate the modes using the data from the burst and plateau cells. This would then allow a more rigorous description of the mixed activity state and can give an estimate of the proportion of bursts and plateaus on average at least.

The authors should also indicate the duration of the recording period that was analyzed to determine pacemaker type. Examination of 1 min of continuous recording would be less than desirable (that is only 1-12 bursts) whereas 10 min or more would be very strong. In any case, we need to be informed on this point.

---

## [Author Response]

*Essential revisions:*

*1) The network model should be removed for the following reasons: i) The model is not needed to provide rationale for the experiments testing for an increased role for synaptic interactions of in rhythm generation. The data themselves suggest this possibility. ii) The model parameters are highly underspecified so it has little (if any) predictive value, nor does it enhance understanding of network organization; and, iii) It is distracting, requiring a lot of effort to understand the model rather than focusing on the relatively simple conclusion that is based on the data (i.e., rhythmic inspiratory synaptic drive to inspiratory neurons increases developmentally).*

A suggested by the reviewers we have removed the entire section on network modeling (Figure 8 and legend has been deleted together with the corresponding text in the Results section). We agree that the network model could be difficult to understand without more parameter information. We hope that this modification now renders the study more focused.

*2) The authors should provide a clear description of how they classified their neurons into the three activity groups (long-lasting plateau-like oscillations with impulses occurring at the beginning of the plateau followed by a depolarization block, short-lasting oscillations and bursts, mixture of long- and short lasting bursts) and mapped these onto biophysical properties. More raw data should be provided of voltage recordings and voltage-clamp records. Details of biophysical measurements should be provided, in tabular form if necessary, on passive properties (input resistance and capacitance) and active currents (current dynamics). Necessary details of the experimental procedures were not always provided, e.g. when FFA and Ril were given sequentially (e.g., Results section, subsection “Membrane conductance underlying the different discharge patterns”: "Ril alone"), how was washout of the first drug assured so the second blocker's effect could be assessed independently? More detailed quantification and statistics are needed for the groupings. For example, when the authors say () "bursting activity of 10 of these was completely blocked" we would like to know the definition and measurement method for a burst or its duration or existence. Moreover, claiming block may seem reasonable from the typical traces provided, but the conclusion remains subjective unless documented quantitatively. While the study deals with the neural system's maturation, embryonic ages of the preps were not always clear. Since this part is fundamental to the rest of study, it could be documented less anecdotally and more quantitatively and rigorously.*

In order to better describe the three different types of pacemaker neurons we recorded in the embryonic preBötzinger network, we now provide data (means ± SEM) and distribution ranges on burst duration, burst amplitude, membrane potential and membrane input resistance (see Figure 2, panel D, and in the text). However, we cannot provide data on individual currents as we never applied specific activating protocols in the presence of selective blockers.

Regarding procedures for FFA and Ril applications, we should make it clear that we never rinsed one drug before applying the other one since, to our knowledge neither of them is fully washable. Therefore, after applying one drug, if we wished to test the independent effects of the second drug, we always set up another experiment using a fresh slice preparation. So, FFA and Ril were either tested alone (as stated in within Results section, subsection “Membrane conductance underlying the different discharge patterns”), or in combination, but never sequentially on the same slice with an intervening wash.

As also requested by referee 2, we now provide greater statistical detail throughout the text for all data groups. In experiments on pacemaker neuron sensitivity to Ril or FFA, we considered bursting activity to be blocked when only single action potentials were generated without any detectable depolarization of the membrane potential. We have completely rearranged Figure 4 in order to more clearly illustrate the finding that under Ril+FFA, only single spikes were produced, but no typical bursting activity (see insets and expanded recording excerpts provided in revised Figure 4).

We agree with the referee that the different developmental stages at which experiments were performed is of a fundamental importance in our study. However, we feel that the initial priority in the report is to convey to the reader the general properties of embryonic pacemaker activities within the E16.5 to E18.5 period. For this reason, the data were pooled for both ages in the first part of the article. The second part of the study (from Figure 5 onwards) then specifically addresses age-dependent differences, and so it is here that we discriminate between data obtained at the two developmental stages. We feel that this organization is quite clearly specified throughout the text.

*3) A significant aspect of the study is based on two sets of observations: 1) imaging and/or patch-clamp demonstrations of burster types and blockers' effects, and 2) blockers' variable effects on the preBötC rhythms recorded locally. These two observations are, however, independent of each other and the causality remains an open question. The cells that were imaged and/or patch-clamped were near the upper surface of the slice where the blockers had ready access. The vast majority of the rhythmogenic population might have been left unaffected, especially in the 450 µm thick slice that can capture a large rostrocaudal substrate in a fetus brainstem. Given, theoretically, 10% of the cross-connected population is sufficient to be pacemakers for the population to produce rhythm (Butera et al. 1999b), a relatively large population unaffected by the blocker may be sufficient to maintain the rhythm such as ones shown in Figure 7. Since preBötC inspiratory pacemaker neurons' contralaterally projecting axons take a rostrally arched course (Koizumi et al. 2013), those pacemakers near the rostral cut-end affected with the blockers might have had lost recurrent connections to a half of the rhythmogenic population (thus loss of their cellular rhythm could have less impact on the rhythm inside slice). The main impact of this diffusion issue in the context of this study is that one of major conclusions is the rhythm at E18.5 is no longer dependent on pacemaker properties. Given that diffusion of drugs through brain tissue decreases as it matures, it is possible that the persistence of rhythm at E18.5 reflects that Ril and FFA did not diffuse through the entire slice. These possibilities should be addressed forthrightly; either by new analysis, a clever but relatively quick set of experiments designed to show that diffusion of drugs is not a problem, or simply qualifying their conclusions with appropriate caveats in Discussion.*

We thank the referee for pointing out a possible issue with the diffusion of FFA and Riluzole applied to 450 µm thick slices at E18.5. It should be noted that they are thinner than slices generally used in similar exogenous drug treatment studies on newborn rodents. Nonetheless, we feel confident that the drugs used in our study are reaching a significant portion of the E18.5 inspiratory network. Firstly, when CNQX or our cocktail of synaptic blockers is superfused, all preBötC rhythmic activity ceases (when monitored either with calcium imaging or macroelectrode electrophysiological recordings). Second, in a direct test for a potential diffusion problem raised by the reviewer, we performed a new series of experiments in which FFA and Riluzole were applied to a thinner (300µm) E18.5 slice preparations. In each case (n=6) we found results closely comparable to those obtained with 450µm thick slices: 1) FFA alone failed to block inspiratory rhythmic activity, and 2), rhythmic activity continued under co-application of Ril + FFA, albeit at a lower frequency than in control conditions (similar to the data presented in the article's Figure 7). These experiments therefore strongly suggest that our reported data showing the persistence of a rhythmogenic capability under Ril + FFA at E18.5 are not simply due to a failure of the drugs to reach deeper lying neuronal populations in our slices.

*4) Over-interpretation of the data is significant concern. This concern can be addressed with revisions to the text, following the minor comments of the reviewers, which are appended. Importantly, in terms of overall significance to understanding mechanisms of rhythm generation it is important to remind readers that these data are based on analysis of the network isolated* in vitro.

We have addressed this concern by making changes to the text according to the specific comments of the reviewers. These modifications are itemized for each comment below.

[Editors' note: further revisions were requested prior to acceptance, as described below.]

1) In their response to the past review the authors state "[…]performed a new series of experiments in which FFA and Riluzole were applied to a thinner (300µm) E18.5 slice preparations." and they claim similar results to 450 µm slice. The authors should include these data in the text, and they should describe the effects of FFA and Riluzole on the thinner slices. These data should be introduced with the caveat about differential diffusion with age. E.g. 'To ensure that reduced sensitivity of rhythm at E18.5 to blockers was not due to reduced diffusion of blockers into the older tissue, we repeated these experiments in thinner slices[…]'

As requested by the referee, the results obtained in our additional experiments on thinner slices have now been included in the Results section. It is now stated: "To ensure that the apparent reduced sensitivity of the network rhythm at E18.5 to blockers was not simply due to a decrease in their diffusion into the older tissue, we repeated these experiments using 300 μm instead of 450 μm slices. However, the finding that the expression of rhythmic activity persisted in such thinner preparations in the presence of FFA + Ril, albeit again at a lower frequency compared to control conditions (51.2 ± 15% ; n = 6; Mann-Whitney test, *P =* 0.02) argues against this possibility".

*2) The authors should provide more clarification about how they classified the three types of cell activities they observed in Figure 2: burst, plateau, mixed. The problem with the current presentation is that there is some overlap in the ranges of Figure 2. The authors should consider plotting their data set of Figure 2 on a burst duration vs. burst amplitude coordinate system, using a different color for each preparation categorized as plateau or burst. This will show the clouds and the overlap and give the skeptical reader a clear idea what is going on. This can be a new figure.*

The mixed cells are somewhat of a problem because average data cannot capture their properties. Qualitatively, the plateaus are easy to distinguish from the bursts; the data are not completely clean, but we see that there is no problem telling the bursts from the plateaus when they occur one without the other in different cells. The problem comes with the mixed cells. Here the verbal and quantitative descriptions and the cloud discrimination break down requested above. This is because average data cannot capture the bimodal nature of the distributions. The text is inadequate on this point because it reports averages and SD for the bimodal data. The authors should recognize explicitly the bimodal nature and try to separate the modes using the data from the burst and plateau cells. This would then allow a more rigorous description of the mixed activity state and can give an estimate of the proportion of bursts and plateaus on average at least.

We thank the referee for these useful comments and suggestions, and we agree that the mixed pattern could be better described. As proposed, we have added a panel (E) in Figure 2 showing a plot of drive potential (DP) duration vs DP amplitude (see reply to comment 3 below) for neurons belonging to the three different groups.

For this analysis we used only cells that had been stably recorded for at least 8 min. It is indeed instructive to show that in the mixed pattern, the DP amplitude/duration relationships of some recorded bursts are similar to those of the purely plateauing pacemakers while other bursts correspond to those of the oscillatory pacemaker subtype. We also accept that giving mean values for overall DP duration and amplitude fails to convey this bimodal nature of the mixed pacemaker discharge pattern. We have therefore now discriminated the two types of bursts and have added the corresponding separate bars (in light green) to the DP amplitude and duration histograms in revised Figure 2D4.

*The authors should also indicate the duration of the recording period that was analyzed to determine pacemaker type. Examination of 1 min of continuous recording would be less than desirable (that is only 1-12 bursts) whereas 10 min or more would be very strong. In any case, we need to be informed on this point.*

As now stated in the Material and Methods section, only neurons recorded in basal conditions for a period of at least 8 minutes were used in our burst parameter analyses. In fact the sub-groups of neurons used to construct the graph in the new panel E of Figure 2 were comprised of cells that were all recorded for 8 minutes or more (also stated in the Figure 2 legend).